# Improved Group Robustness via Classifier Retraining on Independent Splits

**Thien Hang Nguyen**                                                    *nguyen.thien@northeastern.edu*
*Northeastern University, Boston, MA*

**Hongyang R. Zhang**                                                    *ho.zhang@northeastern.edu*
*Northeastern University, Boston, MA*

**Huy Le Nguyen**                                                        *hu.nguyen@northeastern.edu*
*Northeastern University, Boston, MA*

**Reviewed on OpenReview:** *https://openreview.net/forum?id=KgfFAI9f3E*

## Abstract

Deep neural networks trained by minimizing the average risk can achieve strong average performance. Still, their performance for a subgroup may degrade if the subgroup is underrepresented in the overall data population. Group distributionally robust optimization (Sagawa et al., 2020a), or group DRO in short, is a widely used baseline for learning models with strong worst-group performance. We note that this method requires group labels for every example at training time and can overfit to small groups, requiring strong regularization. Given a limited amount of group labels at training time, Just Train Twice (Liu et al., 2021), or JTT in short, is a two-stage method that infers a pseudo group label for every unlabeled example first, then applies group DRO based on the inferred group labels. The inference process is also sensitive to overfitting, sometimes involving additional hyperparameters. This paper designs a simple method based on the idea of classifier retraining on independent splits of the training data. We find that using a novel sample-splitting procedure achieves robust worst-group performance in the fine-tuning step. When evaluated on benchmark image and text classification tasks, our approach consistently performs favorably to group DRO, JTT, and other strong baselines when either group labels are available during training or are only given in validation sets. Importantly, our method only relies on a single hyperparameter, which adjusts the fraction of labels used for training feature extractors vs. training classification layers. We justify the rationale of our splitting scheme with a generalization-bound analysis of the worst-group loss.

## 1 Introduction

Deep neural networks are usually developed with examples in test sets that follow the same distribution as the training set. The performance of deep networks worsens when the test set distribution differs from the training set distribution. This problem has been studied by various literature in the name of out-of-distribution (OOD) generalization. This is crucial in safety-critical applications such as self-driving cars (Filos et al., 2020) and medical image classification (Oakden-Rayner et al., 2020). Tackling distribution shifts for out-of-distribution generalization is one of the most important problems for the real-world deployment of deep learning models.

A notable setting where distribution shifts occur is the group-shift setting, where different data groups may have a distribution shift (Sagawa et al., 2020a). In this setting, there are predefined attributes that divide the input space into different groups of interest. Here, the goal is to find a model that performs well across several predefined groups. Prior work has observed that deep networks learned by empirical risk minimization suffer from poor worst-group performance despite good average-group performance.

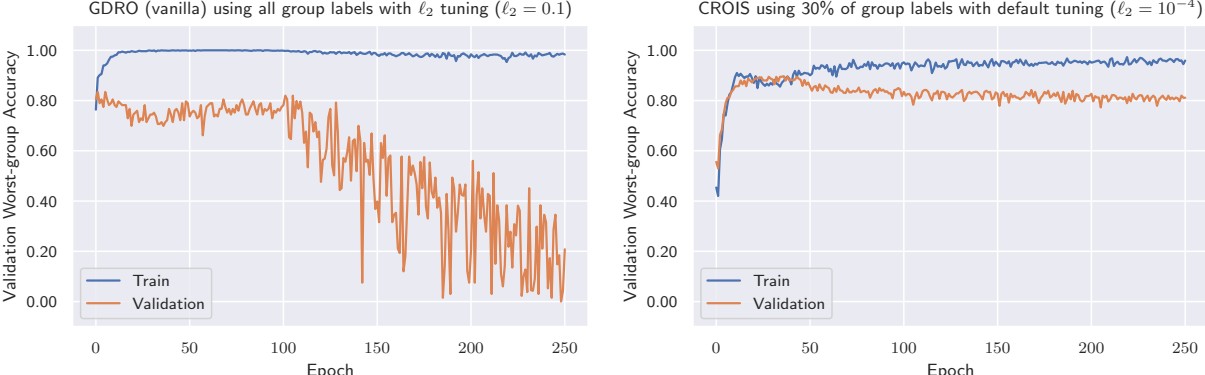

Figure 1: Worst-group learning curve on the Waterbird dataset between group DRO (**left**) and our method (**right**) from the setting in Section 4.2. In the left figure, the validation accuracy of group DRO becomes unstable as the number of epochs increases beyond 100 while the training accuracy remains close to 100%. Our approach instead uses 30% of the training data with group labels for fine-tuning the classification layer of a deep network that is obtained via minimizing the average risk using the rest of the training data without group labels (see also Algorithm 1). This allows our method to reuse features learned in the first stage while improving generalization compared with group DRO.

The difficulty with learning group robust deep networks can be attributed to the phenomenon of shortcut learning (Geirhos et al., 2020) or spurious correlation (Sagawa et al., 2020a; Arjovsky et al., 2019). Shortcut learning poses that minimizing the empirical risk favors models that discriminate based on simpler, spurious features of the data. However, one would like the learning algorithm to produce a model that uses features and correlations that perform well not only on the train distribution but also on all potential distributions that a task may generate, like that of a worst-group distribution.

In recent years, the group-shift setting has received considerable attention. Sagawa et al. (2020a) investigates distributional robust optimization (Ben-Tal et al., 2013) in this setting and introduces group distributionally robust optimization to optimize for the worst-group error directly. Since then, this approach has been widely used for training group-robust models, where it produces strong results that many follow-up works have used as a common baseline (for example, see Liu et al. (2021); Nam et al. (2020); Zhang et al. (2022) and references therein). However, the worst-group error minimization problem is sensitive to small groups (Sagawa et al., 2020a) and requires group labels for all examples at training time.

Followup works in the group shift setting have considered methods to reduce the amount of group labels needed (Liu et al., 2021; Creager et al., 2021; Zhang et al., 2022; Nam et al., 2022). These methods usually follow the framework of first inferring pseudo-group labels using a referenced model (pseudo-labeling) and then applying a group-robust algorithm like minimizing the worst-group loss on the pseudo-labeled data. These methods show promising results, sometimes on par with methods with access to group labels during training. The caveat is that methods in this space usually involve additional hyperparameters, such as those from contrastive learning (Zhang et al., 2022) and semi-supervised learning (Nam et al., 2022). These are nontrivial complexities added to the group DRO procedure. The purpose of our paper is to investigate *whether it is possible to develop group robust models with as few group labels as possible while alleviating the need for expensive parameter turning.*

**Our Contributions.** We answer the above question in the positive by designing a simple approach called Classifier Retraining on Independent Splits or CROIS in short. We replace the pseudo-labeling phase to instead use group labels only for fitting the final classifier layer. Our method achieves good robust performance without relying on various hyperparameters and parameter tuning. We note that this concern has been voiced by other researchers in the community with the goal of prioritizing simple, reproducible research over complex methods (Gulrajani & Lopez-Paz, 2021).

Our method takes advantage of the good features learned by empirical risk minimization (Kang et al., 2019; Menon et al., 2021b) while overcoming the deficiency of its memorization behavior (Sagawa et al., 2020b). We utilize the training data as two independent splits: one group-unlabeled split to train the feature extractor and one group-labeled split to retrain only the classifier with a robust algorithm like group DRO. We say good features for a certain task to mean that there exists a linear classifier utilizing the deep network's features that perform well on our desired task, where features refer to the inputs to the deep network's final linear layer. We demonstrate through ablation studies that using independent splits is crucial for robust classifier retraining. Furthermore, our method's use of group DRO to only a low-capacity linear layer reduces group DRO's sensitivity towards small groups as well as the amount of data needed for group DRO to generalize well. For empirical evidence, see Figure 1 and Figure 3.

For various benchmark data sets whose group labels are only partially given during training, we show strong experimental results on Waterbird, CelebA, MultiNLI, and CivilComments, which improved upon existing methods, including Just Train Twice (Liu et al., 2021) and Spread Spurious Attribute (Nam et al., 2022). The highlight of our method is that we only involve a single parameter (to determine data splitting fractions), and we completely eliminate the pseudo-labeling stage. We demonstrate a surprising result where using only a fraction of the group labels during training, our method shows competitive performance to group DRO that runs on fully-labeled groups. Our results reinforce several recent works showing that deep networks contain good features on image and text classification tasks (Menon et al., 2021b; Kirichenko et al., 2022).

The simplicity of our procedure also allows us to cast it into a formal learning theory framework naturally. We state such a setting and develop a simple generalization bound on the worst-group loss. This result provides some justification for the hyperparameter $p$, which is used to balance the data size for feature learning and the rest for classifier retraining.

The rest of this paper is organized as follows. In Section 2, we will discuss the related works. In Section 3, we will describe the design of our method. In Section 4, we present our experimental results. In Section 5, we provide a generalization bound for the worst-group loss using standard Rademacher complexity techniques. Lastly, we conclude the paper in Section 6. Appendix A provides additional details to support our experimental results. Appendix B states the proof for our theoretical claims.

## 2  Related Work

There are three main settings for the group-shift problem: (1) full availability of group labels, (2) limited availability of group labels, and (3) no availability of group labels, all referring to the training stage. Other related areas include domain generalization and long-tailed classification.

**Fully-labeled group labels during training.** Most methods here revolve around up-weighing minority groups, subsampling minority groups (Sagawa et al., 2020b), or performing group DRO (Sagawa et al., 2020a). Follow-up works include integrating data augmentation via generative model or selective augmentation (Yao et al., 2022) to a robust training pipeline.

**Partially-labeled group labels during training.** In this setting, the approach of inferring more group labels for the group-unlabeled data remains the most popular. These pseudo-group labels are usually generated by training a referenced model that performs the labeling. For example, Liu et al. (2021) utilizes a low-capacity model that creates groups by labeling whether an example is correctly classified by the referenced model or not. Similarly, works like (Creager et al., 2021; Dagaev et al., 2021; Krueger et al., 2021; Nam et al., 2022; 2020) are variants of this approach of inferring pseudo group labels. These methods then proceed to use a group robust algorithm like group DRO (Sagawa et al., 2020a) or Invariant Risk Minimization (Arjovsky et al., 2019) to retrain new deep nets with the newly generated pseudo group labels.

**No group labels during training.** This setting removes the ability to validate knowledge of potential groups. This makes the problem more difficult as it is unclear which correlation to look for during training. Some theoretical works in this space include Lahoti et al. (2020). Sohoni et al. (2020) proposes a popular empirical approach in this setting and has popularized the pseudo-labeling and retraining approach. This setting is related to domain generalization. Gulrajani & Lopez-Paz (2021) shows through mass-scale experiments that

most out-of-distribution generalization methods do not improve over empirical risk minimization given the same amount of tuning and model selection criterion.

**Long-tailed classification.** The long-tailed problem concerns certain classes having significantly fewer training examples than others (see, for example, Zhang et al. (2021) for a survey). Yang et al. (2021) uses random matrix theory to obtain intriguing insights into learning from imbalanced classes, such as the non-monotonicity of adding source data on transfer. Some techniques from the long-tail literature, like margin adjustment and distillation, have been applied to the group-shift setting to account for the group imbalances (Sagawa et al., 2020a; Lukasik et al., 2021; Kini et al., 2021). Li et al. (2023a;b) recently proposed a task modeling and boosting framework to aggregate multiple learned models to counteract the class imbalance problem.

**Representation learning in deep networks.** Investigating the power of the features of deep nets has been of great interest in the long-tail setting (Liu et al., 2019; Menon et al., 2021a; Kang et al., 2019) as well as in the group-shift setting (Menon et al., 2021b). Kang et al. (2019) is one of the first works to provide extensive evidence for the hypothesis that deep nets contain good features via extensive experiments on several long-tailed vision datasets, where different strategies for obtaining feature extractors and fine-tuning the classification layer are examined. There, an ERM-trained feature extractor combined with a non-parametric method of rescaling[1] the classifier layer achieves (then) state-of-the-art results on all three datasets, showing evidence that the features of deep nets can be used to distinguish between rare and frequent classes. These insights are central to the development of our method.

**Memorization in deep learning.** It has now been well known of high capacity deep nets' ability to memorize training examples (Zhang et al., 2017). In the group-shift setting, this behavior has been investigated by Sagawa et al. (2020b), which provides empirical and theoretical justifications for deep nets' memorization behavior of minority groups' training examples. This memorization behavior has also been observed in the other settings, including data imbalances (Feldman & Zhang, 2020), noisy labels (Ju et al., 2022), and fine-tuning pretrained models (Ju et al., 2023).

We would like to point out that, developed concurrently with our work, is a paper by Kirichenko et al. (2022), where the authors similarly discover that classifier retraining on independent splits via a similar procedure improves group robustness. While the main idea of retraining the classifier using independent splits is similar, Kirichenko et al. (2022) focuses more on exploring the features learned by deep networks. In contrast, our work focuses more on controlling model capacity for group DRO, where we limit its use to only the final linear layer. Thus, we believe that our method is of independent interest.

## 3 Method

This section describes the design of our method. First, we lay out the problem setup and the motivation behind this problem. Then, we describe our approach, which involves splitting the dataset into independent splits to conduct training features and classifiers separately.

### 3.1 Preliminaries

For a classification task $\mathcal{T}$ of predicting labels in $\mathcal{Y}$ from inputs in $\mathcal{X}$, we are given training examples $\{(x_i, y_i)\}_{i=1}^n$ that are drawn independent samples from some train distribution $\mathcal{D}_{\text{train}}$. In the domain generalization setting, we want good performance on some unknown test distribution $\mathcal{D}_{\text{test}}$ that is different but related to $\mathcal{D}_{\text{train}}$ through the task $\mathcal{T}$. More explicitly, we wish to find a classifier $f$ from some hypothesis space $\mathcal{F}$ using $\mathcal{D}_{\text{train}}$ such that the classification error $L(f) = \mathbb{E}_{x,y \sim \mathcal{D}_{\text{test}}}[f(x) \neq y]$ of $f$ w.r.t. $\mathcal{D}_{\text{test}}$ is low.

In the group-shift setting (Sagawa et al., 2020a), we further assume that associated with each data point $x$ is an *attribute* $a(x)$ (some sub-property or statistics of $x$) from a set of possible attributes $\mathcal{A}$. These attributes, along with the labels, form the set of possible groups $\mathcal{G} = \mathcal{A} \times \mathcal{Y}$ that each example can take. We denote an input $x$'s group label as $g(x) \in \mathcal{G}$. We then define the classification error of a predictor $f$ (w.r.t. a fixed implicit distribution) restricted to a group $g \in \mathcal{G}$ to be $L_g(f) := \mathbb{E}_{x,y|g(x)=g}[f(x) \neq y]$. The notion of

---

[1]Rescaling each row of the linear classifier using the row's norm to some power. See Kang et al. (2019).

---

**Algorithm 1** Classifier Retraining on Independent Splits (CROIS)

---

**Input:** Training data $D_L$ with group labels and training data without group labels $D_U$. Classifier retraining algorithm $\mathcal{R}$ (default to group DRO). Optional splitting parameter $p$ (default to 1).

1: *Obtain validation sets* by partitioning $D_L$ into $D'_L$ and $D_L^{(val)}$ and $D_U$ into $D'_U$ and $D_U^{(val)}$.

2: (Optional) *Add more unlabeled data* via split proportion $p$: Partition $D'_L$ into two parts, $D_1$ and $D_2$ such that $|D_1| = (1-p) \cdot |D'_L|$ and $|D_2| = p \cdot |D'_L|$. Set $D'_L \leftarrow D_2$ and $D'_U \leftarrow D'_U \cup D_1$.

3: *Obtain the initial model* $f$ by running empirical risk minimization on $D'_U$ and selecting the best model in terms of average accuracy on $D_L^{(val)} \cup D_U^{(val)}$.

4: *Perform classifier retraining* $\mathcal{R}$ with feature extractor $f$ on $D'_L$ and then select the best model in terms of worst-group accuracy on $D_L^{(val)}$ as the final output.

---

*worst-group error* upper bounds the error of $f$ w.r.t. any group $L_{wg}(f) := \max_{g \in \mathcal{G}} L_g(f)$. Using this notation, the group-shift problem aims to discover a classifier in $\arg\min_{f \in \mathcal{F}} \{L_{wg}(f)\} = \arg\min_{f \in \mathcal{F}} \{\max_{g \in \mathcal{G}} L_g(f)\}$. We observe that the group-shift problem is just a particular case of the domain generalization problem when $\mathcal{D}_{\text{test}}$ is the distribution consisting of only the points $(x, y)$ with $g(x)$ being restricted to the worst-group of $f$ in $\mathcal{G}$. Here, group distributional robust optimization solves this objective by performing a minimax optimization procedure that alternates between the model's weight and the relaxed weights of the groups.

**Spurious correlations and memorization.** As an example, consider the Waterbird dataset (Sagawa et al., 2020a), where it has been constructed by combining images of water/land birds from the CUB dataset (Welinder et al., 2010) with water/land backgrounds from the PLACE dataset (Zhou et al., 2017). The task is to distinguish whether an image of a bird is a waterbird or a land bird. Regarding our problem, the type of bird forms the labels $\mathcal{Y}$, and the backgrounds are set to be the attribute $\mathcal{A}$ for each type of bird. Altogether, these form four groups: $\mathcal{G} = \mathcal{Y} \times \mathcal{A} = \{\text{waterbird, landbird}\} \times \{\text{water, land}\}$.

This dataset is constructed so that the proportion of birds on matching backgrounds is significantly more than those of mismatched backgrounds. This is so that the backgrounds could be spuriously correlated with the labels, as predicting the background alone would achieve a high average accuracy w.r.t. the train distribution already. As expected, for models trained by empirical risk minimization, the groups with the highest error are the minority groups where the background mismatches the type of the bird, suggesting that the model is predicting using the background instead of the bird. Furthermore, the fact that these high-capacity models achieve *zero* training error leads to the conclusion that these models not only utilize spurious features like the background to make their predictions but also must have memorized the minority groups during its training process (Sagawa et al., 2020b). These problems are common when there is data imbalance in overparametrized networks (Feldman & Zhang, 2020; Li & Zhang, 2021) or when there is label noise (Ju et al., 2022). In the next section, we propose a method to circumvent these issues.

### 3.2 Our approach

Algorithm 1 presents an outline for our main method: *Classifier Retraining On Independent Splits*, or *CROIS* in short. Given group-labeled data and group-unlabeled data, our method involves several steps:

1. Organize the data into one *group-labeled* split $D'_L$ and one *group-unlabeled* split $D'_U$.

2. Obtain a feature extractor trained by empirical risk minimization with the group-unlabeled split $D'_U$.

3. Perform robust classifier retraining with the group-labeled split $D'_L$, where classifier retraining refers to fine-tuning the final linear layer of a deep network.

In the setting where group labels are limited (as in Section 4.1), $|D_L|$ is much smaller than $|D_U|$, and we do not need to set $p < 1$. There, we primarily concern with partitioning $D_L$ into $D'_L$ and $D_L^{(val)}$. On the other hand, when group labels are available for a large portion of the training dataset (as in Section 4.2) and $|D_U|$ is much smaller than $|D_L|$, the optional parameter $p$ in step 2 controls the size of $D'_U$ to obtain a feature extractor and the number of group labels $D'_L$ used at train time.

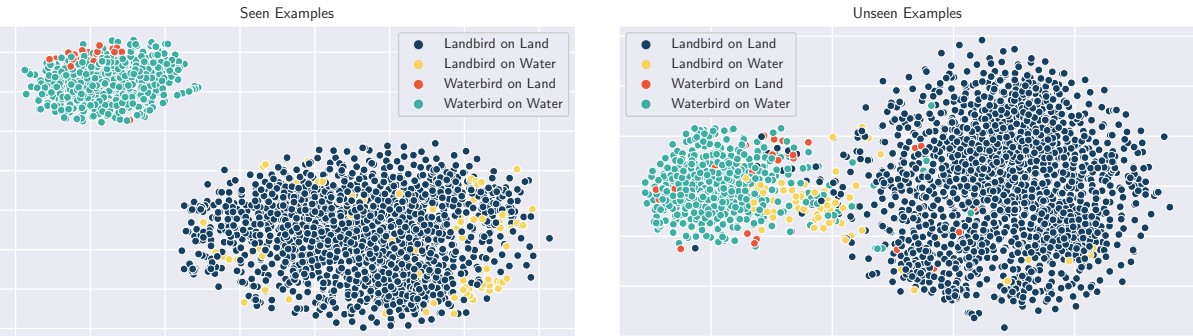

Figure 2: tSNE projection (Van der Maaten & Hinton, 2008) of the features of ResNet-50 on *seen* (**left**) versus *unseen* (**right**) examples from Waterbird.[2] The features of the minority groups (orange and yellow) from the *unseen examples* appear better separated from the majority groups than that of the seen examples. Using unseen examples plays a major role in our method's ability to improve worst-group performance via robust classifier retraining.

**Good features of deep nets.** As discussed in the related works section, there is extensive empirical evidence that deep nets trained by ERM contain features that can distinguish between the minority classes from the majority classes in both the long-tailed setting and the group-shift setting. This suggests that a key to the group-shift problem is correcting the classifier layer, which forms the basis for the first phase of our method. The most efficient way to utilize data for fine-tuning the final layer was left open, and our work focuses on exploring this aspect in more depth.

**Fine-tuning the classifier layer and utilizing group information.** There is a range of possible strategies to utilize group labels. At one extreme, one can rescale using only minimal information, such as the group sizes, or at the extreme, one can maximally utilize group information by training the whole network with group DRO. Our work explores the space between these extremes by limiting the use of group labels to only fine-tune the final layer. Recall that in Kang et al. (2019), rescaling the classifier works best, whereas intuition suggests a data-dependent method like classifier retraining would work better. We hypothesize that this is related to the next issue of our discussion.

**Memorization behavior of deep nets.** As discussed in the related works section, there is strong evidence for deep nets memorizing training examples, which is often believed to be one of the main causes of poor robust performance. One way to circumvent memorization is to control the model's capacity by incorporating some combinations of high $\ell_2$ regularization, early stopping, and other correctional parameters as has been done in Sagawa et al. (2020a). However, methods that are sensitive to different hyperparameter configurations with excessive tuning are not desirable, sometimes leading to reproducibility concerns (Gulrajani & Lopez-Paz, 2021). Our method, instead, does not require excessive tuning. It also extends to numerous settings depending on the availability of group labels.

Tackling this memorization problem is crucial, and we achieve this using independent splits. As memorized examples' (i.e., already correctly classified) loss must be low, their gradients contain little helpful information. Furthermore, the features of memorized examples might not represent their group during test time: Figure 2 presents a visualization of the features between seen versus unseen examples. Thus, combining this observation with the evidence for deep nets containing good features, our method performs robust classifier retraining on *unseen* examples ($D'_L$ in Algorithm 1) in the hope of learning a classifier that utilizes features more representative of examples during test time. Our experimental results confirm these intuitions: the results from Table 3 show that robust classifier retraining without independent split indeed performs worse. Finally, as a side benefit, the independent split lends itself to theoretical analysis, which we provide in Section 5. This further supports the soundness of our method.

---

[2]Half of the data is used to obtain a feature extractor while the other half is used to obtain the features of the unseen examples.

## 4 Experiments

We conduct experiments in two settings: (1) where group labels are only available from the validation split of the datasets (as in Liu et al. (2021); Nam et al. (2022)); and (2) when a fraction of group labels is available from the training split, and all group labels are available from the validation split. Our implementation in PyTorch can be found at https://github.com/timmytonga/crois.

**Setup.** We use a similar setup to Liu et al. (2021) and Sagawa et al. (2020a). To demonstrate the ease of tuning of our method, unless noted otherwise (e.g., Table 2 and parameter $p$ in Table 3), we *fix* the hyperparameters of both the empirical risk minimization (ERM) and the robust classifier retraining phase, reusing standard parameters for ERM (see Appendix A for full hyperparameters and model details). Further results of our method with tuned hyperparameters are presented in Section A.6 of the Appendix.

**Datasets.** We experiment on four datasets:

- **Waterbird** (Sagawa et al., 2020a). Combining the bird images from the CUB dataset (Welinder et al., 2010) with water or land backgrounds from the PLACES dataset (Zhou et al., 2017), the task is to classify whether an image contains a *landbird* or a *waterbird* without confounding with the background. There are 4795 total training examples, whereas the minority group (*waterbird, land background*) has only 56 examples. We report the weighted test *average accuracy* due to the skewed nature of the val and test sets to be consistent with Sagawa et al. (2020a).

- **CelebA** (Liu et al., 2015) is a popular image dataset of celebrity faces. The task is to classify the celebrity in the image is *blond* or *not blond*, with *male* or *not male* as the confounding attribute. There are 162770 total training examples, and the smallest group (*blond, male*) has 1387 examples.

- **MultiNLI** (Williams et al., 2017) is a natural language inference dataset for determining whether a sentence's hypothesis is *entailed* by, is *neutral* with, or *contradicts* its premise. The spurious attribute is negation words like *no, never*, or *nothing*. This task has 6 groups, with 206175 total training and 1521 in the minority group examples (*is entailed* and *contains negation*).

- **CivilComments-WILDS** (Koh et al., 2021) is a natural language dataset where the task is to classify whether a sentence is *toxic* or *non-toxic*. There are 8 demographics – *male, female, white, black, LGBTQ, Muslim, Christian,* and *other religion*– forming 16 groups that *overlap* because a comment can contain multiple demographics. Following Koh et al. (2021), we evaluate all 16 groups but only use the attribute *black* along with the label in training. There is a total of 269038 training examples with 1045 minority examples from (*other religion, toxic*).

### 4.1 Result with validation group labels

**Setup.** In this section, we consider the setting where group labels $D_L$ are available only from the standard validation split, where these group labels can be used for training (Nam et al., 2022) or model selection (Liu et al., 2021). Here, the training split is treated as the group-unlabeled set $D_U$. Most methods in this setting employ some pseudo-labeling approach to generate pseudo group labels that are then used to train a *new* network via a robust algorithm like group DRO. On the other hand, our method simply uses half of $D_L$ for classifier retraining $D'_L$ and the other half for model selection $D_L^{(val)}$ and does not rely on pseudo-labeling. Our method also reuses the initial model for the retraining phase, making our method closer to that of a single-phase procedure with additional fine-tuning.

**Results.** In Table 1, we compare our method against JTT (Liu et al., 2021) and SSA (Nam et al., 2022), where we report the mean and one standard deviation of the test average (*Avg Acc*) and worst-group Accuracy (*Wg Acc*) across 3 random seeds. There, our method outperforms JTT on all 4 datasets and SSA on 3 datasets using default parameters. Note that, unlike our method and SSA, JTT only uses available group labels for model selection. However, JTT requires training many models across two phases, which can be expensive. Furthermore, JTT's model selection can be quite sensitive (see Section 5.4 of Liu et al. (2021)). SSA alleviates this problem of JTT by more efficiently utilizing group labels to infer pseudo-labeling. Finally, our method dispenses altogether with pseudo-labeling while still achieving competitive performance.

Table 1: Experimental results for the setting when only group labels from the validation split are used. Results for JTT (Just Train Twice) and SSA (Spread Spurious Attributes) are taken from Liu et al. (2021) and Nam et al. (2022), respectively. The numbers in parentheses denote one standard deviation from the mean across 3 random seeds. See Table A.7 in Appendix 18 for comparison with additional baselines.

| Method | Waterbird | | CelebA | | MultiNLI | | CivilComments | |
|---|---|---|---|---|---|---|---|---|
| | Avg Acc | Wg Acc | Avg Acc | Wg Acc | Avg Acc | Wg Acc | Avg Acc | Wg Acc |
| JTT | 93.9 | 86.7 | 88.0 | 88.1 | 78.6 | 72.6 | 91.1 | 69.3 |
| SSA | 92.2 (0.87) | 89.0 (0.55) | 92.8 (0.11) | **89.8** (1.28) | 79.9 (0.87) | 76.6 (0.66) | 88.2 (1.95) | 69.9 (2.02) |
| CROIS (ours) | 92.1 (0.29) | **90.9** (0.12) | 91.6 (0.61) | 88.5 (0.87) | 81.4 (0.06) | **77.4** (1.21) | 90.6 (0.20) | **70.3** (0.34) |

Table 2: Worst-group test accuracy for partial group labels from the validation split. Results for SSA and JTT are taken from Table 3 of Nam et al. (2022). The standard deviation is reported based on three independent runs. See Table 19 in Appendix A.7 for comparison with additional baselines.

| % of group-labels from the validation split | CelebA | | | Waterbird | | |
|---|---|---|---|---|---|---|
| | 20% | 10% | 5% | 20% | 10% | 5% |
| JTT (Liu et al., 2021) | 81.1 | 81.1 | 82.2 | 84.0 | 86.9 | 76.0 |
| SSA (Nam et al., 2022) | 88.9 | **90.0** | 86.7 | 88.9 | **88.9** | 87.1 |
| CROIS's Wg Acc | **89.6** (0.4) | 87.6 (0.6) | **87.3** (1.0) | **90.4** (1.0) | 88.2 (0.9) | **87.8** (1.3) |
| CROIS's Avg Acc | 90.8 (0.2) | 91.6 (0.3) | 87.8 (1.6) | 92.4 (0.5) | 93.0 (0.7) | 88.7 (1.6) |

**Discussion.** We clarify the difference between JTT and our method. First, the initial phase of JTT is for *inferring* pseudo-group-labels for the group-unlabeled data. This phase requires careful hyperparameter tuning and capacity control using the group-labeled validation set to accurately produce pseudo group labels (as noted in section 5.4 of Liu et al. (2021)). On the other hand, our method trains a *single* model and simply retrains the last layer with any available group labels. Second, JTT's final performance is limited by group DRO's performance on the full network, which can worsen by mislabeled pseudo labels from the first phase. In contrast, we demonstrate in Section 4.2 that, by limiting group DRO to only the last layer, our method is competitive to full group DRO even when using only a fraction of group labels and minimal tuning.

Compared with SSA (Nam et al., 2022), our method does not rely on pseudo labeling. In SSA, there is a pseudo-labeling phase along with a robust training phase using the inferred group labels. In the first phase, SSA trains a separate network that *predicts the group* rather than the class. By treating the pseudo-labeling problem as semi-supervised learning, SSA's pseudo-labeling capability improves upon JTT. Our results show that our method outperforms SSA on 3 out of 4 datasets while reusing default parameters.

**Reducing validation split size.** Following the setup in JTT (Liu et al., 2021) and SSA (Nam et al., 2022), we vary the size of the validation split (20%, 10% and 5% of the original) to test whether our results still hold in these settings. We consider both the Waterbird and CelebA datasets. Note that for this setting, our method must be additionally *tuned* to account for the increased difficulty of the reduced group-labels quantity. Nevertheless, the smaller examples quantity, along with just training the last layer, makes the extra tuning less expensive (details and setup in Section A.8). We present our results (along with error bars) in Table 2, where our method outperforms JTT and SSA on various percentage levels.

## 4.2 Result with partial training group labels

Next, we consider the setting where group labels are available from both the training split and the validation split. In contrast to Section 4.1, the standard validation split is used *only* for model selection $D_L^{(val)}$ and not for classifier retraining $D_L'$ here. We compare our method using some fraction of the training split's group

Table 3: Comparison between our method CROIS and group DRO. NCRT refers to naive classifier retraining, i.e., when an independent split is *not used* during the retraining phase. Results marked with $^\dagger$ are taken from (Sagawa et al., 2020a). *For Waterbird, we omit the result for $p = 0.05$ due to the small dataset size and inability to sample any minority-group example for robust retraining.

| Method | Waterbird | | CelebA | | MultiNLI | | CivilComments | |
|---|---|---|---|---|---|---|---|---|
| | Avg Acc | Wg Acc | Avg Acc | Wg Acc | Avg Acc | Wg Acc | Avg Acc | Wg Acc |
| ERM | 96.9 | 69.8 | 95.6 | 44.4 | 82.8 | 66.0 | 92.1 | 63.2 |
| GDRO | $93.2^\dagger$ | $86.0^\dagger$ | $91.8^\dagger$ | $88.3^\dagger$ | $81.4^\dagger$ | $77.7^\dagger$ | 89.6 (0.23) | 70.5 (2.10) |
| CROIS' $p$ – group-labeled fraction used for retraining (with $1 - p$ unlabeled fraction for the ERM phase) | | | | | | | | |
| 0.05 | * | * | 91.9 (0.50) | 88.9 (1.10) | 81.8 (0.15) | 73.8 (1.54) | 90.8 (0.40) | 63.3 (7.82) |
| 0.10 | 95.4 (1.10) | 83.5 (3.24) | 91.3 (0.36) | 90.3 (0.82) | 80.8 (0.51) | 75.3 (2.06) | 89.5 (1.81) | 68.7 (1.72) |
| 0.30 | 90.8 (0.35) | **89.6** (1.15) | 91.3 (0.44) | **90.6** (0.95) | 80.0 (0.31) | **77.9 (0.17)** | 89.7 (0.33) | 68.6 (1.53) |
| 0.50 | 90.4 (0.95) | 89.5 (0.59) | 91.9 (0.35) | 88.2 (2.10) | 79.8 (0.26) | 74.4 (1.00) | 89.5 (0.70) | **71.0** (1.50) |
| NCRT | 96.5 | 75.2 | 93.9 | 69.2 | 82.3 | 67.9 | 90.3 | 67.6 |

labels against group DRO using *all* the group labels. Again, we fix the parameters of our method to its standard empirical risk minimization parameter to demonstrate its ease of tuning (see Appendix A).

**Setup.** We study our method with different amounts of training group labels determined by the parameter $p$. This means that $(1 - p)$ fraction of the training split is used to obtain a feature extractor in the first phase $D'_U$ (that uses no group label), and the rest $p$ fraction of group labels are used for robust classifier retraining $D'_L$. This setup allows examining the trade-off between the quality of the feature extractor versus the amount of data available to perform classifier retraining. Additionally, to demonstrate the importance of retraining with *unseen* examples, we experiment with robust classifier retraining using the same data from the first phase, i.e., *without* independent splits – denoted as *NCRT* in the table.

**The parameter $p$.** In practice, we expect that $|D_L|$ is a lot smaller than $|D_U|$, as in Section 4.1. There, Table 2 suggests that reasonable robust performance can be achieved with a small fraction of group labels. In this setup, however, since the amount of group labels is abundant ($D_U = \emptyset$ and $D_L$ is large), we treat $p$ as a tune-able parameter that controls the size of $D'_U$ and $D'_L$. Furthermore, using a $p$ fraction of the available group labels simulates obtaining group labels for a random fraction of the data if there is a budget constraint on group labels.

**Results.** In Table 3, our method outperforms group DRO on both image data sets and yields competitive performance to group DRO on the two text data sets when using only a fraction of group labels and reusing default hyperparameters. Our result implies that comparable or even better robust performance than group DRO can be obtained by collecting group labels for roughly 30% of the available training data (modulo validation). One exception is severe group imbalance cases, as in CivilComments (the minority group consists of only 0.4% of the dataset). There, a higher fraction of group-labeled data is beneficial to obtain more minority-group examples. Hence, a more efficient sampling method to include more minority examples (e.g., filter by labels first) would be beneficial in practice. Finally, the results for naive classifier retraining also show the importance of using an independent split for classifier retraining.[3]

**Trade-off between feature extractor and amount of group-labeled data for robust retraining.** From the results across the data sets, allocating more examples towards training the feature extractor (lower $p$) generally yields higher on-average accuracies. The worst-group error after classifier retraining has a more complex interaction with $p$, as it depends on both the quality of the feature extractor and the amount of group-labeled examples available to perform classifier retraining. While varying the proportion $p$ in our

---

[3]In Sagawa et al. (2020a), group adjustment is observed to improve Waterbird's worst-group accuracy to 90.5%. We also notice an improvement when incorporated here and obtain a $90.3\% \pm 0.62$ test worst-group accuracy. We also observe that similarly to Sagawa et al. (2020a), the adjustment only works for Waterbird but not for CelebA nor MultiNLI.

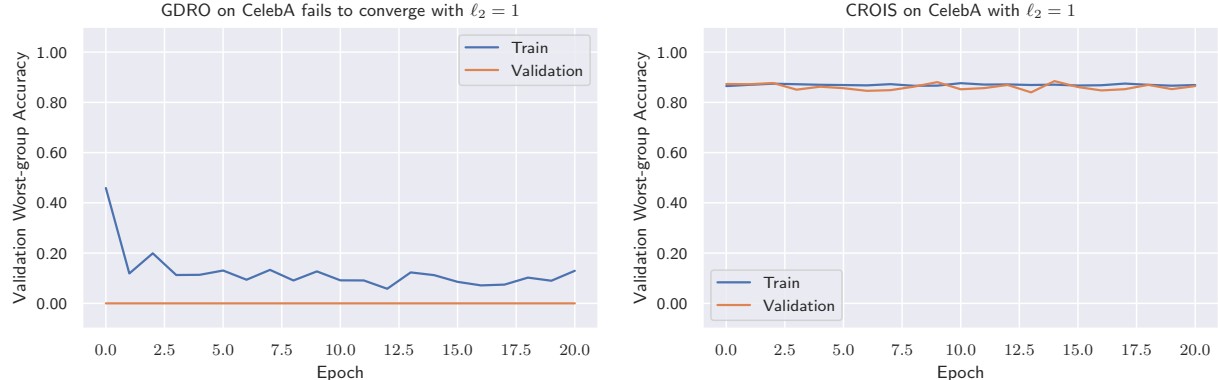

Figure 3: While group DRO often requires high $\ell_2$ regularization to avoid overfitting, setting $\ell_2$ too high might cause instability in group DRO's minimax optimization procedure (**left**).

Table 4: Comparison between reweighting, subsampling, and group DRO as classifier retraining algorithms on top of the same feature extractor ($p = 0.30$).

| Retraining Method | Waterbird | | CelebA | | MultiNLI | | CivilComments | |
|---|---|---|---|---|---|---|---|---|
| | Avg Acc | Wg Acc | Avg Acc | Wg Acc | Avg Acc | Wg Acc | Avg Acc | Wg Acc |
| Reweighting | 95.2 | 87.1 | 92.1 | 85.0 | 78.9 | 67.0 | 88.3 | 56.4 |
| Subsampling | 95.8 | 81.1 | 91.6 | 86.1 | 78.6 | 64.0 | 91.8 | 59.3 |
| GDRO | 91.4 | **90.2** | 91.6 | **90.4** | 80.3 | **78.0** | 89.7 | **69.1** |

experiments gives a rough estimate of this tradeoff, we hypothesize that the availability of minority group examples is the most important for obtaining a robust classifier. We further support this intuition with an ablation study in Section A.5 where removing non-minority examples has an insignificant impact on the final group-robust performance.

**Alleviating group DRO's sensitivity towards model capacity.** Group DRO's requirement for model capacity control via either $\ell_2$ regularization or early stopping is well noted in the literature (Sagawa et al., 2020a). In Table 17, we compare group DRO and our method's sensitivity towards different $\ell_2$ regularization. While our method's performance on Waterbird is relatively uniform, group DRO is more sensitive to different $\ell_2$ settings on CelebA. When $\ell_2 = 1$ for CelebA, group DRO fails altogether (see Figure 3). On the other hand, our method achieves consistent performance across different $\ell_2$ settings. Our method controls the model capacity by limiting group DRO to only the last layer. This alleviates group DRO's tendency to overfit and simplifies parameter tuning (as in Figure 1 for Waterbird).

### 4.3 Ablation studies

**Obtaining a good feature extractor.** An ablation study on the effects of different validation accuracy and initial algorithms on the feature extractor's quality (measured by robust performance after classifier retraining) is presented in Section A.4. Similarly to previous works Kang et al. (2019), empirical risk minimization provides the best features over reweighting or group DRO (both requiring group labels). We find a positive correlation between validation average accuracy and features' quality. This then serves as a proxy for our method's model selection criterion in the first phase, significantly simplifying parameter tuning over other two-phase methods in the group-shift setting.

**Group DRO is better than reweighting and subsampling for classifier retraining.** Table 4 contains results on using different classifier-retraining methods. We observe that group DRO produces the best group-

robust performance (since group DRO is designed for this setting, after all). Reweighting and subsampling seem effective on the vision datasets but fail to perform on the NLP datasets.

Table 5: Comparison between retraining with group DRO on the full network (*Full*) versus last layer retraining (*LL*).

| GDRO $p = 0.30$ | Waterbird | | CelebA | | MultiNLI | | CivilComments | |
|---|---|---|---|---|---|---|---|---|
| | Avg Acc | Wg Acc | Avg Acc | Wg Acc | Avg Acc | Wg Acc | Avg Acc | Wg Acc |
| LL (CROIS) | 91.4 | **90.2** | 91.6 | **90.4** | 80.3 | **78.0** | 89.7 | 69.1 |
| Full | 90.5 | 79.8 | 91.6 | 78.3 | 80.8 | 75.1 | 90.4 | 69.1 |

**Classifier retraining outperforms full retraining with group DRO.** Classifier retraining plays a central role in our method. In Table 5, we compare fine-tuning with group DRO on the full DNN versus just the last layer for an independent split of $p = 0.30$. We see that group DRO on the last layer is much better than full group DRO on most datasets (except CivilComments). However, the main difference is that while the last layer retraining requires little additional tuning, we must search for different regularization strengths for group DRO when applied to the full network. our method can be shown to be quite robust to different parameter settings and regularization strengths (details in Appendix A.6).

**Deep nets learned by minimizing the average risk contain good features.** The positive result for our decoupled training procedure provides strong evidence for deep nets containing good features for the group-shift problem. While this is consistent with findings in the literature on vision datasets (Kang et al., 2019; Menon et al., 2021b), our work further provides some of the first evidence of this hypothesis in non-vision tasks, where the same result would not have been possible without independent split, as evident in the result for naive classifier retraining in Table 3.

**Simplified model selection.** The model selection criterion of picking the best average validation accuracy model simplifies hyperparameter tuning compared to other two-phase methods. This decision has been chosen mainly from the ablation experiments in Section A.4 of the Appendix, where we observe that higher average validation accuracy generally suggests better features.

## 5 Theoretical Justification of Data Splitting with Generalization Bounds

In this section, we complement our empirical results with Rademacher complexity-based generalization bounds for both phases of our method: one for the standard average loss for the deep net's feature extractor and another for the worst-group generalization bound for linear classifiers. These serve as an explanation for balancing the right proportion of samples within both phases, especially when the data imbalance between subgroups is significant. We provide an upper bound on the worst-group loss that depends on the worst-group sample size. We then argue that the average generalization error lower bounds the worst-group generalization error, and a balanced splitting of the samples is beneficial to control the worst-group error sufficiently.

**Generalization bound on average losses.** We first present the standard generalization bound for deep nets here. Given a dataset $S := \{(x_1, y_1), \ldots, (x_n, y_n)\}$, consider a function class $\mathcal{F}$ consists of $L$-layers feedforward neural network with $\rho$-Lipschitz activations along with the composed function class with the dataset $\mathcal{F}_{|S}$:

$$\mathcal{F} := \left\{ x \to \sigma_L \left( W_L \sigma_{L-1} \left( \cdots \sigma_1 \left( W_1 x \right) \cdots \right) \right) \mid \left\| W_i^T \right\|_{1,\infty} \leq B \right\}$$

$$\mathcal{F}_{|S} := \left\{ \left( \ell \left( f(x_1), y_1 \right), \ldots, \ell \left( f(x_n), y_n \right) \right) \mid f \in \mathcal{F} \right\}.$$

If we collect samples $x_1, \ldots, x_n$ into rows of $X \in \mathcal{X}^{n \times d}$ and if the activations $\sigma_i$ are $\rho$-Lipschitz with $\sigma_i(0) = 0$, then we can show the following generalization bound for $\mathcal{F}$:

**Proposition 1** (See, e.g., Telgarsky (2021))**.** *Suppose that $f \in [a, b]$ for all $f \in \mathcal{F}$ for some finite $a < b$. Then with probability at least $1 - \delta$*

$$\sup_{f \in \mathcal{F}} \mathbb{E}\left[f(Z)\right] - \frac{1}{n} \sum_{i=1}^{n} f(z_i) \leq \frac{2}{n} \|X\|_{2,\infty} (2\rho B)^L \sqrt{2 \ln(d)} + 3(b - a) \sqrt{\frac{\ln(2/\delta)}{2n}}.$$

We defer the proof to Appendix B. We can view a simple convolution layer at depth $l$ with kernels $K_1^{(l)}, \ldots, K_{k_l}^{(l)}$ as linear layers of Toeplitz matrices $W_1^{(l)}, \ldots, W_{k_l}^{(l)}$ representation the kernel. Then, for the $(1, \infty)$-norm for a convolution net with $L$ layers, we scale with the size of the kernels rather than the dimension with standard linear layers:

$$\left\|W_j^{(l)}\right\|_{1,\infty} = \sum_i \left|K_{j,i}^{(l)}\right|.$$

With this view, we can apply the bound above to convolutional networks as in our experiments.

**Generalization bound for worst-group losses.** Next, we derive a generalization bound for the worst-group loss for linear classifiers. Suppose we are working with a binary classification problem with $G$ groups, where for a sample $x \in \mathcal{X}$, we have that $g(x) \in [G]$. Consider the hypothesis class of linear classifiers with bounded norm i.e.

$$\mathcal{H} := \left\{w \in \mathbb{R}^d \mid \|w\| \leq B\right\}.$$

Since the 0-1 loss $\ell(x, y) = 1[x \neq y]$ is a bit hard to work with, we instead consider the logistic loss

$$\ell(x, y) = \ln\left(1 + \exp\left(-yx\right)\right).$$

Hence, the function class along with our dataset $S := \{(x_1, y_1), \ldots, (x_n, y_n)\}$ we are considering for generalization is

$$\mathcal{F} := \left\{(x, y) \rightarrow \ell\left(w^T x, y\right) \mid \|w\| \leq B\right\}$$
$$\mathcal{F}_{|S} := \left\{\left(\ell\left(w^T x_1, y_1\right), \ldots, \ell\left(w^T x_n, y_n\right)\right) \mid \|w\| \leq B\right\}.$$

We can further partition the dataset into its $G$ groups: $S_1, S_2, \ldots, S_G$, where $S_i := \{(x, y) \in S \mid g(x) = i\}$ with sizes $n_i$ each. Let

$$\hat{R}_i(f) := \frac{1}{n_i} \sum_{(x_j, y_j) \in S_i} \ell(f(x_j), y_j), \qquad R_i(f) := \mathbb{E}_{(X,Y)|g(X)=i}\left[\ell(f(X), Y)\right]$$

denote the empirical and population group loss of $f$. We can then show the following result:

**Theorem 2.** *If $\mathcal{F}$ is a class of bounded linear classifiers and $l$ is the logistic loss, then for the binary classification with $G$ groups, we have that with probability at least $1 - \delta$:*

$$\sup_{f \in \mathcal{F}} \underbrace{\max_{i \in G} R_i(f) - \hat{R}_i(f)}_{\text{WG generalization error}} \leq \max_{i \in G} \frac{2}{n_i} B \cdot \left\|X^{(i)}\right\|_F + 3\left(\ln(2) + B\right) \sqrt{\frac{\ln(2G/\delta)}{2n_i}}.$$

We defer the proof to Appendix B.

**Two stage training and data balancing: Empirical validation.** Note that since the worst-group loss upper bounds the average loss, the average loss is a natural lower bound on the worst-group loss. The generalization gap for the worst-group loss is not only influenced by its hypothesis class and the amount of data per group but it is also influenced indirectly by the lower bound given by the average loss for training the feature extractor. Hence, this suggests that splitting the data into different proportions (via the parameter $p$) across two phases in our method is a means to trade-off between the worst-group and average

Table 6: Generalization gap between different splitting proportions of the training data for both phases.

| | CelebA | | | Waterbird | | |
|---|---|---|---|---|---|---|
| Splitting proportion $p$ | 0.1 | 0.3 | 0.5 | 0.1 | 0.3 | 0.5 |
| Test worst-group loss | 0.287 | 0.275 | 0.312 | 0.389 | 0.27 | 0.313 |
| Train worst-group loss | 0.249 | 0.269 | 0.236 | 0.183 | 0.25 | 0.295 |
| Generalization gap | 0.038 | **0.006** | 0.076 | 0.206 | **0.02** | 0.018 |

loss generalization gap. Note that, as with most existing generalization bounds for deep networks, these bounds are most likely not informative and are just suggestions for algorithms design.

We present some supporting empirical evidence in Table 6 showing the worst-group loss and average loss generalization gap for various splitting proportions $p$ for CelebA and Waterbird. The generalization gap shown below is selected from the best validation epoch. There, we obtain the best generalization when the splitting is "balanced" ($p = 0.30$) as in our other experiments.

## 6 Conclusion

In this paper, we propose classifier retraining on independent splits as a simple method to reduce the number of group annotations needed for improving worst-group performance as well as alleviate group DRO's requirement for careful control of model capacity (Sagawa et al., 2020a). Our experimental results show the effectiveness of our method on four standard datasets across two settings and provide evidence that deep nets contain good features for the group-shift problem.

**Future Work.** The richness of deep net features can potentially be helpful in solving the seemingly harder group-agnostic setting (where no group label is available) by allowing the practitioner to focus on obtaining a robust classifier given a feature extractor, where we have shown that reasonable robustness can be achieved with relatively few group-labels, which makes the problem seem closer in reach. On a broader note, while most work in representation learning focuses on producing good features (either with supervised, unsupervised, or self-supervised approaches), further examinations into different ways to perform classifier retraining in different settings (as in our work) could give a fuller picture to the features quality of different methods.

**Broader Impact Statement.** Worst group robustness is closely related to fairness in AI, where issues like biases of machine learning models are considered (Hardt et al., 2016; Ding et al., 2021). Our method seeks to improve the group robustness of deep learning models, which is important as machine learning models become more ubiquitous.

## Acknowledgement

Thanks to Pavel Izmailov and Michael Zhang for several discussions related to this paper. T. N. is partially supported by a seed/proof-of-concept grant from the Khoury College of Computer Sciences, Northeastern University.

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

# A Experimental Details

## A.1 Infrastructure

We performed our experiments on 2 PCs with one NVIDIA RTX3070 and one NVIDIA RTX3090. Our implementation is built on top of the code base from Liu et al. (2021). Experimental data is collected with the help of Weights and Biases (Biewald, 2020).

## A.2 Models

We use ResNet50 (He et al., 2016) with ImageNet initialization and batch-normalization for CelebA and Waterbird. We use pretrained BERT (Devlin et al., 2018) for MultiNLI and CivilComments. We use the original train-val-test split in all the datasets and report the *test* results. Cross-entropy is used as the base loss for all objectives. SGD with momentum (set to 0.9) is used for the vision datasets while the AdamW optimizer with dropout and a fixed linearly-decaying learning rate is used for BERT. We use a batch size of 16 for CivilComments and 32 for the rest of the datasets. We do *not* use any additional data augmentation or learning rate scheduler in our results.

## A.3 Hyperparameters

Table 7 contains the hyperparameters used in our experiments in Sections 4.2 and 4.1. Note that these are the standard parameters for obtaining an ERM model for these datasets as in previous works (Sagawa et al., 2020a; Liu et al., 2021). The only difference is that we train Waterbird and CelebA for slightly shorter epoch due to finding no further increase in validation accuracies after those epochs.

In our experiments, unless noted, we do not tune for any other hyperparameters. For the second phase of CivilComments, we do not use the default regularization but opt for 0 since the linear layer already has low capacity. However, adding further regularization does not seem to have much of an effect as in section A.6.

Table 7: Hyperparameters used in the experiments. The slash indicates the parameters used in the first phase (feature extractor) versus the second phase (classifier retraining).

|  | Waterbird | CelebA | MultiNLI | CivilComments |
|---|---|---|---|---|
| Learning Rate | $10^{-4}/10^{-4}$ | $10^{-4}/10^{-4}$ | $2 \times 10^{-5}/2 \times 10^{-5}$ | $10^{-5}/10^{-5}$ |
| $\ell_2$ Regularization | $10^{-4}/10^{-4}$ | $10^{-4}/10^{-4}$ | $0/0$ | $10^{-2}/0$ |
| Number of Epochs | 250/250 | 20/20 | 20/20 | 6/6 |

## A.4 Ablation studies: Obtaining a good feature extractor

In this section, we examine the different factors that can potentially impact the quality of the feature extractor.

**Impact of The Feature Extractor's Algorithms.** We provide evidence that ERM-trained models produce the best features for worst-group robustness. We conduct an experiment on Waterbird, where instead of using ERM to obtain a feature extractor, we perform group DRO and Reweighing instead in the first phase. The results are presented in Table 8. While using reweighing or group DRO for the first phase defeats the purpose of reducing the number of group labels needed (whereas ERM doesn't need any), it is informative to examine the features alone. There, we see that even though ERM does not use group labels, it provides the best features for robust classifier retraining on an independent split.

**Impact of Early Stopping and Validation Accuracies on The Feature Extractor.** We present an ablation study of how different early stopping epoch (Figure 4), average validation accuracy (Figure 5 left), and worst-group accuracy (Figure 5 right) of the initial ERM trained model affect the group DRO clsasifier

Table 8: Effects of different methods for obtaining a feature extractor on test average accuracy and test worst-group accuracy (with ResNet50 on Waterbird).

| Feature extractor via | Test Avg Acc | Test Wg Acc |
| --- | --- | --- |
| Reweighing | 90.1 | 88.8 |
| Group DRO | 90.8 | 88.6 |
| ERM | 90.5 | **90.2** |

retraining phase of CROIS. The results here are from performing CROIS with $p = 0.30$ on Waterbird across a wide variety of epochs. Table 9 presents the full data generated for this section.

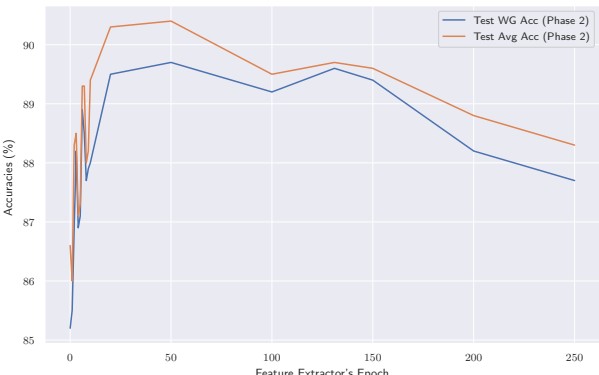

Figure 4: The effect of using different epochs for the feature extractor (phase 1) on classifier retraining's (phase 2) test accuracies.

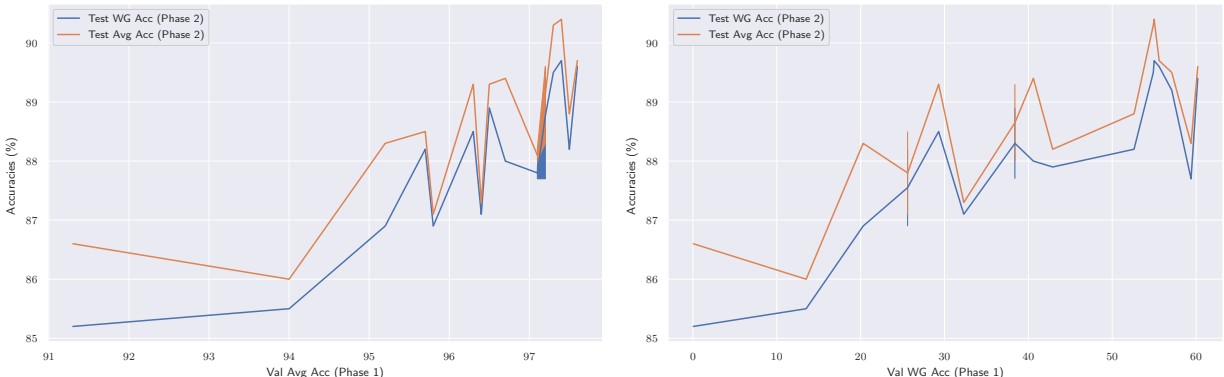

Figure 5: The effect of different *validation average accuracies* (**left**) and *validation worst-group accuracies* (**right**) from the feature extractor (phase 1) on classifier retraining's (phase 2) test accuracies.

### A.5 Further studies on robust classifier retraining

**Impact of robust retraining on independent splits.** In this section, we examine how robust retraining affects the model's prediction of $D_U$ and $D_L$ before and after robust classifier retraining on independent split (with $p = 0.30$). Tables 10 and 11 show the accuracy on $D_U$ and $D_L$ for Waterbird and CelebA. The "Points changed" column indicates the number of points that the model changes prediction after robust retraining per group along with the total number of examples in that group (with percentage in parentheses). The worst group is underlined in the tables.

Table 9: CROIS with group DRO ($p = 0.30$) on Waterbird. Average (*Avg Acc*) and Worst-group (*Wg Acc*) Accuracies for various epochs of the feature extractor ("Phase 1") and the corresponding test accuracies for classifier retraining ("Phase 2"). While training for longer epochs seems to help with average and worst-group accuracy for phase 2, the benefit is small. Hence, simply selecting the best validation average accuracy model, row Epoch 131 and denoted *BEST* here, yields good enough features that simplify our training procedure and model selection criteria.

| CROIS ($p = 0.3$) | Feature Extractor (Phase 1) | | Classifier Retraining (Phase 2) | | | |
|---|---|---|---|---|---|---|
| Phase 1 Epoch | Val Avg | Val WG | Val Avg | Val WG | Test Avg | Test WG |
| 0 | 91.3 | 0.05 | 87.8 | 85.6 | 86.6 | 85.2 |
| 1 | 94 | 13.5 | 88 | 87.6 | 86 | 85.5 |
| 2 | 95.2 | 20.3 | 89.6 | 88 | 88.3 | 86.9 |
| 3 | 95.7 | 25.6 | 90.1 | 88 | 88.5 | 88.2 |
| 4 | 95.8 | 25.6 | 88.9 | 88.2 | 87.1 | 86.9 |
| 5 | 96.4 | 32.3 | 89.2 | 88.7 | 87.3 | 87.1 |
| 6 | 96.5 | 38.4 | 90.5 | 88.7 | 89.3 | 88.9 |
| 7 | 96.3 | 29.3 | 90.6 | 88.7 | 89.3 | 88.5 |
| 8 | 97.1 | 38.4 | 90.1 | 89.3 | 88 | 87.7 |
| 9 | 97.1 | 42.9 | 90.4 | 89.9 | 88.2 | 87.9 |
| 10 | 96.7 | 40.6 | 91.2 | 90.2 | 89.4 | 88 |
| 20 | 97.3 | 54.9 | 91.4 | 91 | 90.3 | 89.5 |
| 50 | 97.4 | 55 | **91.5** | 91 | **90.4** | **89.7** |
| 100 | 97.2 | 57.1 | 90.5 | 90.2 | 89.5 | 89.2 |
| 131 (Best) | **97.6** | 55.6 | 90.9 | 90.2 | 89.7 | **89.6** |
| 150 | 97.2 | **60.2** | 90.7 | 90.2 | 89.6 | 89.4 |
| 200 | 97.5 | 52.6 | 91 | 90.2 | 88.8 | 88.2 |
| 250 | 97.2 | 59.4 | 91 | **90.4** | 88.3 | 87.7 |

Table 10: Model's prediction of $D_U$ and $D_L$ on Waterbird *before* and *after* robust classifier retraining on independent split. Before denotes the ERM training phase. The worst-group is underlined.

| Waterbird ($p = 0.3$) | Accuracy on $D_U$ | | | Accuracy on $D_L$ | | |
|---|---|---|---|---|---|---|
| | Before | After | Points changed | Before | After | Points changed |
| Avg Acc | 100 | 94.5 | 184/3356 (5.48%) | 96.4 | 89.6 | 162/1439 (11.3%) |
| Group 0 (73.0%) | 100 | 92.6 | 180/2430 (7.41%) | 99.6 | 88.2 | 122/1068 (11.4%) |
| Group 1 (3.84%) | 100 | 99.3 | 1/141 (7.09%) | 76.7 | 100 | 10/43 (23.3%) |
| Group 2 (1.17%) | 100 | 100 | 0/38 (0.00%) | 44.4 | 100 | 10/18 (55.6%) |
| Group 3 (22.0%) | 100 | 99.6 | 3/747 (0.40%) | 90.8 | 92.3 | 20/310 (6.45%) |

Table 11: Model's prediction of $D_U$ and $D_L$ on CelebA *before* and *after* robust classifier retraining on independent split. Before denotes the ERM training phase. The worst-group is underlined.

| CelebA ($p = 0.30$) | Accuracy on $D_U$ | | | Accuracy on $D_L$ | | |
|---|---|---|---|---|---|---|
| | Before | After | Points changed | Before | After | Points changed |
| Avg Acc | 96.5 | 92.4 | 7537/113939 (6.61%) | 95.6 | 92.0 | 3274/48831 (11.3%) |
| Group 0 (43.7%) | 96.7 | 91.7 | 2609/50311 (5.19%) | 95.8 | 91.3 | 1011/21318 (4.74%) |
| Group 1 (41.4%) | 99.6 | 92.2 | 3489/46652 (7.48%) | 99.6 | 92.2 | 1493/20222 (7.38%) |
| Group 2 (14.1%) | 89.9 | 95.2 | 1000/16012 (6.25%) | 86.8 | 93.7 | 517/6868 (7.53%) |
| Group 3 (0.87%) | 46.3 | 91.8 | 439/964 (45.4%) | 35.5 | 95.3 | 253/423 (59.8%) |

Interestingly, after robust retraining, the worst group almost always switches from the minority group to the majority group regardless of the data split.

**The importance of minority Examples: subsampled retraining.** As alluded to in Section 4.2, group imbalance seems to play an important role in the robust performance of CROIS. To further demonstrate this point, we consider how CROIS performs when the second phase is subsampled versus when it is allowed additional non-minority examples.

Table 12: Performance of CROIS when retraining is on a subsampled split versus the full split. Subsampling the split doesn't seem to impact CROIS's performance, indicating the importance of the availability of minority examples.

| Dataset (minority group size, fraction) | Subsampled retraining | | Full retraining | |
|---|---|---|---|---|
| | Avg Acc | Wg Acc | Avg Acc | Wg Acc |
| CelebA ($n = 423$, 0.87%) | 90.6 | 89.7 | 91.9 | 88.9 |
| Waterbird ($n = 18$, 1.2%) | 91.3 | 87.6 | 90.3 | 88.9 |

As the result in Table 12 shows, there isn't a significant difference between the two sampling strategies, suggesting that the availability of minority group examples plays an important role in robust classifier retraining.

## A.6 Hyperparameter tuning: CROIS vs. group DRO

In this section, we examine in more depth CROIS's sensitivity to hyperparameter tuning in comparison to group DRO.

**Further hyperameter exploration on CROIS.** In the main body, we have demonstrated CROIS's effectiveness even with just using the same hyperparameters to train an ERM model. Here, we present results for further additional parameter tuning on the robust classifier retraining phase. These results provide empirical evidence for CROIS's potential as well as robustness to different hyperparameter settings.

$\ell_2$ **regularization.** We investigate whether *additional* regularization would be helpful to classifier retraining with group DRO on CelebA (Table 13) and Waterbird. (Table 14) We further examine the effects of regularization on CivilComments (Table 15) to support our choice in Section A. The results in the tables contain the mean and 1 standard deviation across 3 random seeds.

Table 13: Effects of $\ell_2$ regularization on classifier retraining with group DRO on CelebA.

| $\ell_2$ Reg. | $p = 0.50$ | | $p = 0.30$ | | $p = 0.10$ | |
|---|---|---|---|---|---|---|
| | Avg Acc | Wg Acc | Avg Acc | Wg Acc | Avg Acc | Wg Acc |
| 1 | 91.6 (0.06) | 87.4 (2.76) | 90.9 (1.13) | 88.4 (0.78) | 91.2 (0.35) | 90.0 (0.70) |
| $10^{-2}$ | 92.1 (0.32) | 87.6 (1.56) | **91.8** (0.21) | 87.5 (3.54) | 92.0 (0.44) | 88.3 (2.39) |
| $10^{-4}$ | 91.9 (0.35) | **88.2** (2.10) | 91.3 (0.44) | **90.6** (0.95) | 91.3 (0.36) | **90.3** (0.82) |
| 0 | **92.2** (0.25) | 86.8 (2.30) | 91.5 (0.07) | 87.8 (3.96) | **92.1** (0.47) | 88.1 (2.73) |

Table 14: Effects of $\ell_2$ regularization on classifier retraining with group DRO on Waterbird.

| $\ell_2$ Reg. | $p = 0.50$ | | $p = 0.30$ | | $p = 0.10$ | |
|---|---|---|---|---|---|---|
| | Avg Acc | Wg Acc | Avg Acc | Wg Acc | Avg Acc | Wg Acc |
| 1 | 89.6 (0.93) | 88.9 (1.37) | **91.7** (0.44) | **89.8** (0.68) | 94.5 (0.50) | 85.8 (0.15) |
| $10^{-2}$ | **90.4** (1.31) | 89.3 (0.61) | 90.6 (1.25) | 89.2 (1.31) | 95.1 (0.40) | 86.3 (0.83) |
| $10^{-4}$ | **90.4** (0.95) | **89.5** (0.59) | 90.8 (0.35) | 89.6 (1.15) | **95.4** (1.10) | 83.5 (3.24) |
| 0 | 89.8 (0.53) | 89.3 (0.70) | 90.6 (1.31) | 89.1 (1.16) | 95.1 (0.35) | **86.4** (0.90) |

Table 15: Effects of $\ell_2$ regularization on classifier retraining with group DRO on CivilComments.

| $\ell_2$ Reg. | $p = 0.50$ | | $p = 0.30$ | | $p = 0.10$ | |
|---|---|---|---|---|---|---|
| | Avg Acc | Wg Acc | Avg Acc | Wg Acc | Avg Acc | Wg Acc |
| 1 | 88.8 (1.30) | 70.0 (1.63) | 89.5 (0.35) | 66.4 (1.99) | 89.3 (1.69) | 68.9 (2.64) |
| $10^{-2}$ | 89.4 (0.99) | 70.6 (0.42) | 89.5 (0.29) | 68.5 (0.87) | 88.6 (1.70) | **70.2** (1.63) |
| 0 | **89.5** (0.70) | **71.0** (1.50) | **89.7** (0.33) | **68.6** (1.53) | **89.5** (1.81) | 68.7 (1.72) |

**Learning rate.** We examine the effects of different learning rates on CROIS on CelebA and Waterbird in Table 16. A lower learning rate seems to be more beneficial.

Table 16: Effects of varying learning rate on CROIS $p = 0.30$ on CelebA and Waterbird. We fix $\ell_2$ regularization to $10^{-4}$.

| | Learning rate | $10^{-5}$ | $10^{-4}$ | $10^{-3}$ | $10^{-2}$ | $10^{-1}$ |
|---|---|---|---|---|---|---|
| CelebA | Average accuracy | **92.2** | 91.4 | 91.2 | 90.1 | 91.2 |
| | Worst-group accuracy | 88.3 | 90 | **90.3** | 87.8 | 82.8 |
| Waterbird | Average accuracy | 89.7 | 90.6 | **94.2** | 93.2 | 93.9 |
| | Worst-group accuracy | **89.5** | 88.9 | 87.1 | 88.8 | 78.5 |

**Sensitivity to model capacity: CROIS versus group DRO.** In this section, we present a comparison between CROIS and group DRO test performance with different $\ell_2$ regularization configurations on CelebA and Waterbird in Table 17. On CelebA, group DRO is quite sensitive to model capacity while it is less so on Waterbird. We note that group DRO fails to converge to a good stationary point when $\ell_2 = 1$ on CelebA (see Figure 3).

Table 17: Worst-group test accuracy for group DRO and CROIS ($p = 0.30$) with different $\ell_2$ regularization.

| $\ell_2$ reg. | CelebA | | | | | | Waterbird | | | | | |
|---|---|---|---|---|---|---|---|---|---|---|---|---|
| | 0 | $10^{-4}$ | $10^{-3}$ | $10^{-3}$ | $10^{-1}$ | 1 | 0 | $10^{-4}$ | $10^{-3}$ | $10^{-3}$ | $10^{-1}$ | 1 |
| GDRO | 81.7 | 81.7 | 81.7 | 83.9 | **87.8** | 0.00 | 86.8 | 86.8 | 86.8 | 86.8 | **87.1** | 86.5 |
| CROIS | 90.6 | **91.5** | 90.0 | 90.0 | 90.3 | 90.0 | 90.3 | **90.6** | 90.6 | 90.6 | 90.0 | 88.2 |

## A.7 Additional comparison to other methods

We provide additional baselines for comparison in the Tables below.

**Additional baselines for group labels from only the validation set.** In Table 18, we compare CROIS against JTT (Liu et al., 2021) and SSA (Nam et al., 2022), as well as additional baselines like CVaR DRO

Table 18: Experimental results for the setting when only group labels from the validation set are used. Results for C-DRO (CVaR DRO), LfF, EIIL, JTT, and SSA are from Nam et al. (2022). Results for UMIX are from Han et al. (2022). Results for CnC are from Zhang et al. (2022). The numbers in parentheses denote one standard deviation from the mean across 3 random seeds.

| Method | Waterbird | | CelebA | | MultiNLI | | CivilComments | |
|--------|-----------|--------|--------|--------|----------|--------|---------------|--------|
| | Avg Acc | Wg Acc | Avg Acc | Wg Acc | Avg Acc | Wg Acc | Avg Acc | Wg Acc |
| C-DRO | 96.0 | 75.9 | 82.5 | 64.4 | 82.0 | 68.0 | 92.5 | 60.5 |
| LfF | 91.2 | 78.0 | 85.1 | 77.2 | 80.8 | 70.2 | 92.5 | 58.8 |
| EIIL | 91.2 | 78.0 | 85.1 | 77.2 | 80.8 | 70.2 | 92.5 | 58.8 |
| JTT | 93.9 | 86.7 | 88.0 | 88.1 | 78.6 | 72.6 | 91.1 | 69.3 |
| UMIX | 93.0 (0.5) | 90.0 (1.1) | 90.1 (0.4) | 85.3 (4.1) | N/A | N/A | 90.6 (0.4) | 70.1 (0.9) |
| CnC | 90.9 (0.1) | 88.5 (0.3) | 89.9 (0.9) | 88.8 (0.9) | N/A | N/A | 81.7 (0.5) | 68.9 (2.1) |
| SSA | 92.2 (0.87) | 89.0 (0.55) | 92.8 (0.11) | **89.8** (1.28) | 79.9 (0.87) | 76.6 (0.66) | 88.2 (1.95) | 69.9 (2.02) |
| CROIS | 92.1 (0.29) | **90.9** (0.12) | 91.6 (0.61) | 88.5 (0.87) | 81.4 (0.06) | **77.4** (1.21) | 90.6 (0.20) | **70.3** (0.34) |

Table 19: Additional comparison where group labels from the training set are available. Results for LISA are taken from Yao et al. (2022). Results for CAMEL are taken from Model Patching.

| Method | Waterbird | | CelebA | | MultiNLI | | CivilComments | |
|--------|-----------|--------|--------|--------|----------|--------|---------------|--------|
| | Avg Acc | Wg Acc | Avg Acc | Wg Acc | Avg Acc | Wg Acc | Avg Acc | Wg Acc |
| ERM | 96.9 | 69.8 | 95.6 | 44.4 | 82.8 | 66.0 | 92.1 | 63.2 |
| GDRO | 93.2$^\dagger$ | 86.0$^\dagger$ | 91.8$^\dagger$ | 88.3$^\dagger$ | 81.4$^\dagger$ | 77.7$^\dagger$ | 89.6 (0.23) | 70.5 (2.10) |
| LISA | 91.8 (0.3) | 89.2 (0.6) | 92.4 (0.4) | 89.3 (1.1) | N/A | N/A | 89.2 (0.9) | **72.6** (0.1) |
| CAMEL | 90.9 (0.9) | 89.1 (0.4) | N/A | N/A | N/A | N/A | N/A | N/A |
| CROIS' $p$ – group-labeled fraction used for retraining (with $1-p$ unlabeled fraction for the ERM phase) | | | | | | | | |
| 0.10 | 95.4 (1.10) | 83.5 (3.24) | 91.3 (0.36) | 90.3 (0.82) | 80.8 (0.51) | 75.3 (2.06) | 89.5 (1.81) | 68.7 (1.72) |
| 0.30 | 90.8 (0.35) | **89.6** (1.15) | 91.3 (0.44) | **90.6** (0.95) | 80.0 (0.31) | **77.9** (0.17) | 89.7 (0.33) | 68.6 (1.53) |

(Levy et al., 2020), LfF (Nam et al., 2020), EIIL (Creager et al., 2021), CnC (Zhang et al., 2022) and UMIX (Han et al., 2022). There, we report the mean and one standard deviation of the Test Average (*Avg Acc*) and Worst-Group Accuracy (*Wg Acc*) across three random seeds.

**Additional baselines for group labels from the training set.** In the setting where group labels are available from the training set, we compare our method against additional baselines like LISA (Yao et al., 2022).

### A.8 Fraction of the validation set implementation details from Section 4.1

Following the setup in Section 4.1 and the setup as in Liu et al. (2021); Nam et al. (2022), we further reduce the validation set to only a small fraction, 5%, 10%, and 20%. We investigate CROIS's performance in this very few group-labels setting across CelebA and Waterbird in Section 4.1. We note that the highly reduced sample size poses a new challenge and makes it harder to simply reuse the default parameters.

- **Tuning $\ell_2$ regularization:** When using so little data, overfitting can become a bigger problem, even when just training a low-capacity linear classifier. Hence, we tune for higher values for $\ell_2$ regularization across $\{10^{-4}, 10^{-2}, 1, 10\}$.

- **Tuning learning rate:** We also tune the learning rate across $\{10^{-5}, 10^{-4}, 10^{-3}, 10^{-2}\}$ instead of simply reusing default parameters.

- **The use of group labels and model selection:** Since the number of examples for classifier retraining is now significantly reduced, it might be wasteful to further split our available group labels for validation.

Instead, we use all the available group labels for robust classifier retraining and perform model selection in the second phase via the *train worst-group accuracy*. The low capacity linear layer and higher $\ell_2$ regularization allow us to avoid overfitting when performing model selection this way. The feature extractor from the first phase is selected via the best average accuracy on the full group-unlabeled validation set.

- **Smaller batch size:** Since group DRO requires group-balanced sampling, a batch size greater than the number of examples in a certain group would cause duplicate sampling of the minority-group examples in the same step, artificially increasing the weight for that group. We further tune for batch sizes across a grid of powers of 2 less than the smallest group or the default batch size (e.g. we search across $\{4, 8, 16\}$ if the size of the smallest group is 17).

In Table 2, we present the results for CROIS with the above modifications and compare them to CROIS and JTT. There, robust retraining for CelebA is performed with an $\ell_2$ regularization of 0.1, batch size of 8, and learning rate $10^{-5}$. For Waterbird, we found that batch size 8, weight decay 1, and learning rate $10^{-5}$ are best for 20% and 10% reduction. For 5% reduction, we further reduce the batch size to 4 (since the minority group only has 7 examples) and increase the weight decay to 10.

The results show that CROIS maintains its robust performance even at greatly reduced group labels. This implies that even a few (minority) examples can help debias the final layer classifier with proper configurations.

## B  Proofs of Theorem 1

*Proof of Theorem 1.* We look to apply the standard Rademacher Complexity generalization bound on the function class $\mathcal{F}_{|S}$. Recall that the standard Rademacher Complexity generalization bound (see for example Theorem 13.1 in Telgarsky (2021) and references therein) gives that for a function class $\mathcal{F}$ where for each $f \in \mathcal{F}$, $f \in [a, b]$, and dataset $S = \{z_1, z_2, \ldots, z_n\}$ with $n$ i.i.d. samples from some population distribution, we have that with probability at least $1 - \delta$

$$\sup_{f \in \mathcal{F}} \mathbb{E}\left[f(Z)\right] - \frac{1}{n} \sum_{i=1}^{n} f(z_i) \leq \frac{2}{n} R(\mathcal{F}_{|S}) + 3\left(b - a\right) \sqrt{\frac{\ln\left(2/\delta\right)}{2n}},$$

where for Rademacher random variables $\epsilon_i \in \{-1, +1\}$, we have that

$$R(\mathcal{F}_{|S}) := \mathbb{E}_{\epsilon}\left[\sup_{f \in \mathcal{F}} \sum_{i=1}^{n} \epsilon_i f(z_i)\right]$$

is the Rademacher complexity of $\mathcal{F}$. Now, we simply bound the Radamacher Complexity of $\mathcal{F}_{|S}$ (Theorem 14.1 of Telgarsky (2021)):

$$R(\mathcal{F}_{|S}) \leq \|X\|_{2,\infty}\left(2\rho B\right)^L \sqrt{2\ln\left(d\right)}.$$

This gives us the generalization bound. $\qquad\square$

*Proof of Theorem 2.* If we collect samples $x_1, \ldots, x_n$ into rows of $X \in \mathcal{X}^{n \times d}$, then Theorem 13.3 from Telgarsky (2021) gives that

$$R\left(\mathcal{F}_{|S}\right) \leq B \cdot \|X\|_F.$$

Similarly, if we collect samples of $S_i$ into rows $X^{(i)} \in \mathcal{X}^{n_i \times d}$, then we have

$$R\left(\mathcal{F}_{|S_i}\right) \leq B \cdot \left\|X^{(i)}\right\|_F.$$

Then combining with the Rademacher-complexity generalization bound gives that with probability at least $1 - \delta$,

$$\sup_{f \in \mathcal{F}} \mathbb{E}\left[f(Z)\right] - \frac{1}{n} \sum_{i=1}^{n} f(z_i) \leq \frac{2}{n} B \cdot \|X\|_F + 3\left(\ln(2) + B\right) \sqrt{\frac{\ln\left(2/\delta\right)}{2n}},$$

where the range of $f \in \mathcal{F}_{|S}$ is $l(\langle w, xy \rangle) \leq \ln(2) + \langle w, xy \rangle \leq \ln(2) + B$. For a group $i \in G$ with training samples $X^{(i)}$ and $f$ being linear classifiers with max norm $B$, we have that with probability at least $1 - \delta$

$$\sup_{f \in \mathcal{F}} \mathbb{E}_{(X,Y)|g(X)=i} \left[ \ell(f(X), Y) \right] - \frac{1}{n_i} \sum_j \ell(f(x_j), y_j) \leq \frac{2}{n_i} B \cdot \left\| X^{(i)} \right\|_F + 3 \left( \ln(2) + B \right) \sqrt{\frac{\ln(2/\delta)}{2n_i}}.$$

Now, letting $\hat{R}_i(f) := \frac{1}{n_i} \sum_j \ell(f(x_j), y_j)$ and $R_i(f) := \mathbb{E}_{(X,Y)|g(X)=i} \left[ \ell(f(X), Y) \right]$ denotes the empirical and population group loss of $f$ and taking a union bound over all the groups, we have that with probability at least $1 - \delta$

$$\max_{i \in G} \sup_{f \in \mathcal{F}} R_i(f) - \hat{R}_i(f) \leq \max_{i \in G} \frac{2}{n_i} B \cdot \left\| X^{(i)} \right\|_F + 3 \left( \ln(2) + B \right) \sqrt{\frac{\ln(2G/\delta)}{2n_i}}.$$

Since the RHS is finite, we can swap the max and sup on the LHS and we have

$$\sup_{f \in \mathcal{F}} \underbrace{\max_{i \in G} R_i(f) - \hat{R}_i(f)}_{\text{WG generalization error}} \leq \max_{i \in G} \frac{2}{n_i} B \cdot \left\| X^{(i)} \right\|_F + 3 \left( \ln(2) + B \right) \sqrt{\frac{\ln(2G/\delta)}{2n_i}}.$$

$\square$

**Implications and takeaways: Comparison to standard pretraining and finetuning.** As mentioned in the related works section as well as during the discussion of the motivation for our method, pretraining and then finetuning is a now well-known and established strategy in many domains. Our work differs the most significantly from this standard strategy through:

1. The use of independent splits: Traditional pretraining and finetuning reuses the dataset for both phase with the possibility of additional labels (contrastive learning, long-tailed learning, etc.). In our paper, we demonstrate through extensive experiments the importance of independent split when performing classifier retraining for .

2. The use of a group robust algorithm for finetuning: We mainly utilize group DRO for the classifier retraining phase. In contrast, most other works utilize strategies like reweighing or subsampling for finetuning. We demonstrated in our experiments that group DRO yields the best robust performance over other methods.

Our work provides evidences that the features of ERM trained DNNs are rich enough to solve the group-shift problem (when an abundant amount of group labels is available to retrain the classifier) and one of the major reasons for poor worst-group performance of an ERM trained DNN is within its classifier layer. We then further demonstrate that even a few group labels can sufficiently "fix" the classifier to achieve better group-robust performance.

This knowledge can potentially be useful towards solving the seemingly much harder group-agnostic setting (where no group label is available) by allowing the practitioner to focus on obtaining a robust classifier given an ERM trained feature extractor. Our experiments further show that reasonable robustness can be achieved with relatively few group-labels (that are not used to obtain the feature extractor), which makes the problem seem closer in reach.

