# OpenReview forum: "Improved Group Robustness via Classifier Retraining on Independent Splits"
_TMLR — Accepted by TMLR_

### Review · Reviewer_kBVv · 2023-05-28

**Summary Of Contributions:**

This paper proposed a sample-splitting method named _classifier retraining on independent splits_ (CROIS) for improving the worst group performance, using both group-labeled and group-unlabeled data. The author aimed to reduce the need for group labels and hyperparameter tuning. The proposed method was evaluated on several datasets.

**Audience:**

Yes

**Broader Impact Concerns:**

The author did not present a Broader Impact Statement.

The worst group performance is closely related to fairness in machine learning. The author should discuss how the proposed method may affect robustness and fairness if used for sensitive data.

**Claims And Evidence:**

No

**Requested Changes:**

- I was surprised that the first sentence was erroneous. Being "independent and identically distributed" is a property of a collection of random variables, not two distributions (unless we are talking about distributions of distributions, which seem not the case here).
- "Similar to other problems in OOD generalization, DNNs learned by ERM...": models similar to problems? (+ grammatical error)
- Group DRO is the standard: please provide support.
- Please write the full name when an acronym is used for the first time.
- "This framework encapsulates many problems" as well as all imaginable supervised learning problems, so it adds no information.

In general, please proofread the manuscript and improve the academic writing.
Please improve the explanation of the motivation of the proposed method.


**Strengths And Weaknesses:**

## Strengths

- This work aims to use as few group labels as possible, which is nice
- The sample-splitting procedure can be combined with other robust training methods
- The proposed method is simple and can be easily implemented

## Weaknesses

- The motivation "ERM-trained models contain good features (?), features of memorized examples are bad, regularization is costly -> independent splits" is not sufficiently explained and thus not very convincing. The soundness of this approach should be clarified.
- There is no assumption on the relationship between the group-labeled split and group-unlabeled split.
- The theoretical analysis seems not related to the proposed method? The analysis provides very limited insights and guarantees.

---

> ### Author Response · Authors · 2023-06-11
> **Revision**
>
> We have uploaded a revised version of our paper that incorporate the requested changes. We thank the reviewer for the suggestions. We summarize our changes below:
>
> - We corrected the first sentence to be more precise.
> - We fixed grammatical errors in some sentences that the reviewer pointed out.
> - We added more references to support that Group DRO is a standard baseline.
> - We added the full name in front of every acronym the first time it is used across multiple sections to improve readability. We also use the full name instead the acronym in several places and cite again the source of the name of the method or acronym to improve readability.
>
> > "This framework encapsulates many problems" as well as all imaginable supervised learning problems, so it adds no information.
> - We agree that the above sentence is quite broad and have removed the sentence.
> - We have also added a broader impact statement that discusses the relationship between our work and other works in areas like fairness.
>
> We thank the reviewer again for the suggestions and are happy to answer any remaining questions.

---

### Review · Reviewer_9Pqs · 2023-06-06

**Summary Of Contributions:**

This paper proposes a simple method named CROIS for learning a model to address the challenge of group-shift settings, i.e., the performance of underrepresented class in the training set might have be low. The motivation of CROIS comes from the fact that DNNs trained by empricial risk minimization have the ability to provide good features despite that might exploit the spurious features. The proposed method is composed of two steps: first train a model without the group labels, and then train using group distributionally robust optimization (GDRO) with the group labels. The experiment results show that the method can improve the performance on the underrepresented group in the training set.





**Audience:**

Yes

**Claims And Evidence:**

Yes

**Requested Changes:**

To better address or classify the concern of using validation set's data for training.

**Strengths And Weaknesses:**

Strengths:

- The paper is well-motivated and well-written.
- The idea is straight forward yet effective.
- The method has theoretical support

Weakness:

- The selection of hyper-parameter, especially $p$, need to be manually selected. It would be better to find a more principled way to pick this hyper-parameter. In my opinion it controls a very important trade-off.
- The proposed method seems to use the validation set's group label in Table 1 and 2. This might be impractical for real world applications as they are not always available. Also it could lead to an unfair comparison with other baseline methods.

---

### Review · Reviewer_MZy7 · 2023-06-13

**Summary Of Contributions:**

This paper presents a simple method to improve the group robustness of a classifier. The main idea is to first train a feature extractor through a conventional ERM procedure, and retrain only the last layer (classification layer) on the group-labeled split. The resulting procedure is much simpler than the existing methods often requiring complicated procedures such as pseudo-group-labeling or tuning sensitive hyperparameters, and does not require tons of group-labeled examples. Through extensive empirical validation, the proposed method (coined as CROIS) is demonstrated to outperform existing methods, and more importantly, to be easier to use, in the sense that it requires less hyperparameter tuning and is less sensitive to the various choices to be accounted for a classifier training for group robustness. The paper also presents a simple generalization bound for the worst group loss.

**Audience:**

Yes

**Broader Impact Concerns:**

The paper does not bring potential broader impact concerns.

**Claims And Evidence:**

Yes

**Requested Changes:**

The argument in section 5.3 is not that convincing. While it is true that the average group loss is a lower-bound on the worst group loss, since it is a lower-bound, it is not clear how reducing the average group loss is related to improving the worst group loss. Moreover, the bound itself does not tell anything about the tradeoff between the splitting proportion $p$ and the worst group loss, nor anything about the quality of the feature extractor. It is quite expected to be honest because the bound itself is not that informative from the first place. I'd like to suggest to tone-down the arguments in section 5.3, or provide more empirical observations to support them; for instance, one may test with varying $p$ values, measure the quality of the feature extractors via average group accuracies, and show the correlation between average group generalization bound and the worst group generalization bound.

**Strengths And Weaknesses:**

Strengths
- The paper is well-written and easy to follow.
- The proposed method is simple and versatile, I like it requires less hyperparameter tuning or training tricks.
- The experimental results are promising.

Weaknesses
- As the authors already admitted in the paper, there is a concurrent work (Kirichenko et al., 2022) presenting a similar work. I'm not sure about the Journal's policy on concurrent work, but I don't want to give a discount due to the concurrent work.
- The theoretical analysis is a bit weak; it is a rather straightforward application of standard theory to a classification problem with groups, and does not seem to provide further insight about the empirical findings.

---

> ### Author Response · Authors · 2023-06-20
> **Author Response**
>
> We thank the reviewer for the detailed and positive comments. We would like to respond to your concerns about our paper below.
>
> **Relation to concurrent paper.** We thank the reviewer for providing detailed comments! Indeed we are aware of the concurrent work by Kirichenko et al. (2022). We note that while the conceptual message of our paper is indeed similar to that paper, our method is still quite different, which we believe would be of independent interest. In particular, we use sample splitting to train the feature representations first, then apply group distributional robust optimization to retrain the last layer, whereas Kirichenko et al. (2022) train the last layer on a reweighted balanced dataset. We also compare different fine-tuning strategies like reweighting in Table 4 but found that group DRO yields the best performance with minimal tuning needed. In addition, we have been communicating with the authors of that paper about the fact that our paper is concurrent and appeared online around the same time.
>
> **Comments regarding the theoretical result.** Regarding Section 5.3, as the reviewer pointed out, the paper is of an empirical nature. The theoretical analysis is presented more as supporting evidence for our method. In particular, due to the simple sample-splitting procedure, we managed to show a generalization bound for our method (directly from standard techniques). We believe this is still interesting independently of the experiments, and we hope that the modeling aspects can serve as a starting point for future work to further investigate this spurious correlations setting.
>
> Recognizing the reviewer’s concern, we have revised the paper to state that our result suggests a trade-off between the worst-group and average loss generalization gap. We also revised the introduction to reflect this change. Lastly, we added some discussions to recognize the limitations of our analysis in Section 5.3.

---

### Decision · Action_Editors · 2023-07-24

**Recommendation:** Accept as is

**Comment:**

The paper proposes a new approach to group robustness that requires minimal group annotation, and has accompanying theoretical guarantees. Three reviewers found the paper to be intuitive, well executed, and to be of sufficient interest to the community.

**Audience:**

Three reviewers agree that the paper tackles an important problem that has received sustained study, and that the proposed method offers a useful addition to this body of work.

**Claims And Evidence:**

Three reviewers agree that the papers' claims are generally well-supported by evidence. There were requests for clarification of certain portions of the original submission, such as the nature of the generalisation bound. These appear to have been adequately addressed during the revision.